# Recent Advancements in Tracking Bacterial Effector Protein Translocation

**DOI:** 10.3390/microorganisms10020260

**Published:** 2022-01-24

**Authors:** Julie Braet, Dominiek Catteeuw, Petra Van Damme

**Affiliations:** iRIP Unit, Laboratory of Microbiology, Department of Biochemistry and Microbiology, Ghent University, K.L. Ledeganckstraat 35, 9000 Ghent, Belgium; Julie.Braet@UGent.be (J.B.); Dominiek.Catteeuw@UGent.be (D.C.)

**Keywords:** effector, FAST, genetic code expansion, localization, NanoLuc, self-labeling enzymes, translocation, type III secretion

## Abstract

Bacteria-host interactions are characterized by the delivery of bacterial virulence factors, i.e., effectors, into host cells where they counteract host immunity and exploit host responses allowing bacterial survival and spreading. These effectors are translocated into host cells by means of dedicated secretion systems such as the type 3 secretion system (T3SS). A comprehensive understanding of effector translocation in a spatio-temporal manner is of critical importance to gain insights into an effector’s mode of action. Various approaches have been developed to understand timing and order of effector translocation, quantities of translocated effectors and their subcellular localization upon translocation into host cells. Recently, the existing toolset has been expanded by newly developed state-of-the art methods to monitor bacterial effector translocation and dynamics. In this review, we elaborate on reported methods and discuss recent advances and shortcomings in this area of tracking bacterial effector translocation.

## 1. Bacterial Effectors: Function and Delivery into Host Cells

Pathogenic and symbiotic bacteria and their hosts are continuously engaged in an evolutionary arms race in which hosts have evolved multiple lines of defenses to cope with infection. In turn, to optimize bacterial survival, replication and dissemination, pathogens evolved a myriad of mechanisms to counteract or deceive host immune surveillance systems and to exploit host responses. In part, this is achieved by the translocation of (proteinaceous) virulence factors–designated as effectors–into host cells by means of dedicated secretion systems. Gram-positive bacteria, such as *Listeria monocytogenes* (*L. monocytogenes*), usually deliver their effectors via general secretion systems, including the conserved Sec secretion system found in all classes of bacteria in which unfolded proteins harboring an N-terminal signal peptide are translocated across the membrane [1]. The Sec system consists of a SecYEG cytoplasmic membrane translocase which forms a transport channel through which the substrate is translocated, a process driven by the ATP-dependent motor protein SecA [2]. In contrast, effector secretion is more challenging for diderm (Gram-negative) bacteria as effectors have to be translocated across two membranes, i.e., the inner and outer membrane, or even three membranes in the context of infection during which effectors are additionally translocated across the host cell membrane. To date, 10 different specialized secretion systems (Type I-X secretion systems, T1SS-T10SS) have been recognized and three of these systems–T3SS, T4SS and T6SS–are known to directly deliver effector proteins into target cells [3,4,5]. Type III and type VI secretion systems are widespread in Gram-negative bacteria, whilst type IV secretion systems can also be found in Gram-positive bacteria [6,7,8]. Most pathogenic bacteria make use of various protein secretion systems to successfully invade their respective host organisms [9], and many Gram-negative pathogens including members of the Enterobacteriaceae family (e.g., *Salmonella, Shigella, Yersinia* and *Citrobacter*) make use of the T3SS, often referred to as the injectisome because of its needle-like structure composed of three major components in case of active injectisomes; a base complex, a needle-like core and a translocon [10]. The base complex, consisting of several rings, spans both the inner and outer bacterial membrane. From this base complex, the needle extends into the extracellular space. As such, unfolded proteins–prevented from folding by chaperones to maintain a secretion-compatible conformation–can pass from the bacterial cytoplasm through the inner hollow core of the needle into the extracellular space or into the host cell [11]. The latter is enabled by the translocon–a structure consisting of translocators which are also substrates of the T3SS–that is assembled onto the translocator assembly platform and inserted into the host cell membrane immediately upon host cell contact [12]. Similar to T3SSs, T4SSs are able to translocate proteins into eukaryotic host cells [13]. Evolutionary, T4SSs are related to the bacterial conjugation system and can deliver single proteins as well as protein-protein or protein-DNA complexes [3]. T6SSs on the other hand share structural homology with bacteriophage tails and are believed to secrete proteins important for virulence into host cells as well as competitor bacteria [3,14].

Once inside the host cell, effector proteins play functionally very diverse roles, all aiming at bacterial survival and proliferation inside the host. For instance, symbiotic nitrogen-fixing *Rhizobium* sp. T3SS effectors (T3Es) are crucial for nitrogen fixation and nodule initiation in host plants, and are in return provided with nutrients from the host [15]. Related to pathogenesis on the other hand, translocation of T3Es into host cells was shown to be imperative for bacterial virulence, as demonstrated, e.g., by the complete loss of virulence in case of T3SS-deficient *Yersinia* species or by the impaired bacterial dissemination of pathogenicity island 2 (SPI-2)-deficient *Salmonella* in their hosts [16,17]. Strikingly, some pathogens secrete tens or even hundreds of effectors (e.g., *Legionella* pneumophila encodes over 300 T4Es, with 3027 predicted protein-encoding genes in total), while others only secrete a few effectors (e.g., *Pseudomonas aeruginosa* encodes 4 T3Es) [18,19]. In case of *Salmonella*, two T3SSs encoded on two different pathogenicity islands, i.e., SPI-1 and SPI-2, facilitate the delivery of around 50 T3Es into infected cells [20,21]. While SPI-1 encoded T3Es direct the early biogenesis of the *Salmonella*-containing vacuole (SCV), SPI-2 T3Es are responsible for the subsequent SCV maturation, intracellular bacterial survival, and direction of the systemic phase of infection [22].

## 2. The Past and Future Ways of Monitoring Bacterial Effector Translocation

Although effectors generally display high levels of sequence diversity, they are usually identified by means of machine-learning approaches based on the presence of protein sequence features including eukaryotic-like domains or features indicative of protein secretion signals (e.g., translocation or localization signals) or (validated) empirically, e.g., by making use of T3SS mutants [23,24]. To determine effector functions, assessment of T3E translocation dynamics is typically performed in both a qualitative and quantitative manner as for instance timing of secretion during infection, co-effector secretion, concentrations of secreted effectors and localizations inside host cells can shed light on an effector’s mode of action. Original studies typically analyzed effector secretion by probing bacterial culture supernatant or lysates of infected cells for the presence of the effectors, as for instance performed for the *Salmonella enterica* subsp. *enterica* serovar Typhimurium (further referred to as *S. typhimurium*) T3Es SipB and SipC by using monoclonal antibodies against these effectors [25]. However, it should be noted that only few effector antibodies have been reported so far, indicating that endogenous epitope tagging with, e.g., FLAG, HA or Myc is usually required as demonstrated for *S. typhimurium* T3Es SopE, SopB and SptP [26]. However, this method is less suited for high-throughput studies and does not allow for real-time monitoring of effector translocation. Consequently, several enzyme-linked or fluorescent reporter methods have been developed aiming to track secretion in a spatio-temporal manner, which have extensively been reviewed in [27,28]. Importantly, the type of translational fusion (N- or C-terminal tagging) of the effectors under study with a reporter has to be carefully considered in function of the type of effectors being studied. For instance, T3Es are typically fused at their C-terminus since secretion signals are usually contained N-terminally, while in case of T4Es N-terminal translational fusions are used since their secretion signal is typically found in the C-terminus [29,30]. In this review, besides a brief recapitulation of these more routine methods, we will elaborate on recent advances and state-of-the-art additions to the toolset of translocation assays. In particular the use of FAST, NanoLuc and self-labeling enzyme tags, besides the use of genetic code expansion to track bacterial effector translocation (and dynamics in real-time), will be discussed in detail (Figure 1 and Table 1).

### 2.1. Fluorescence-Based Methods to Track Bacterial Effector Translocation

Bacterial effector translocation can be studied by several methods providing a fluorescence-based readout. To compare these methods, it is thus imperative to consider several fluorescence properties such as absorption/emission spectra. Besides, fluorescence quantum yields (Φ) defined by the ratio of emitted photons to absorbed photons, and molar absorption coefficients (ε) determined by the capacity of light absorption, jointly determine the fluorescence brightness.

#### 2.1.1. GFP Impairs T3E Translocation; Split-GFP as Possible Alternative

GFP or GFP-like fluorescent proteins (FPs) are universally recognized as important molecular tools to study general protein properties and dynamics. However, while the Sec-dependent secretion system allows translocation of GFP-tagged proteins, folded GFP was shown to block the translocation of GFP-effector fusions through the T3SS precluding its use to monitor bacterial T3E secretion. This inhibitory potential on translocation was observed when assessing the secretion of the endogenously C-terminally eGFP-tagged T3Es SptP and SopE2 in *S. typhimurium* as immunodetection was unable to detect eGFP-tagged SptP or SopE2 in the culture supernatants, whereas native SptP and SopE2 could readily be detected [12,36,45,46]. In spite of the reasonably large size of eGFP (238–239 AA, 27 kDa, 24 Å in diameter) potentially perturbing T3E function (Figure 2a), it was rather the compact and stable fold of GFP that was initially suggested to hinder T3E secretion. Only later, it was argued that the mechanical stability–reflecting the ease of protein unfolding under force–rather than the thermodynamic stable and compact fold of GFP impairs its secretion. More specifically, while GFP was shown to exhibit similar thermodynamic stabilities as SptP and SopE2, SptP and SopE2 displayed a 5 to 6-fold lower unfolding force and are thus mechanically more labile compared to GFP (SptP: 20,5 piconewton (pN); SopE2: 17 pN; GFP: 116 pN) [46].

Alternatively, the use of a split-GFP complementation strategy was devised as a possible solution to study effector translocation into host cells. Viewing the unique β-can protein fold of GFP, in which 11 β-strands form a β-barrel containing an α-helix with the covalently bonded chromophore at the inside [47], separating the 10 N-terminal β-strands (GFP1–10, AA 1–214, 24 kDa) from the 11th β-strand (GFP11, AA 215–230, 1.8 kDa) resulted in the generation of a split-GFP system in which fluorescence is produced only upon complementation of the two GFP moieties (Figure 2a,b) [48]. The GFP1–10 sequence is based on superfolder GFP (sfGFP) containing seven additional mutations improving its fluorescence, complementation and solubility (Figure 2a). Split-GFP complementation was proven successful in monitoring T3E translocation into host plant cells as the localization of *Pseudomonas syringae* AvrB and AvrRps4 T3Es translationally fused to GFP11 could readily be observed in *Arabidopsis thaliana* expressing the GFP1–10 counterpart [32]. The split-GFP reporter system was also used to study the localization and dynamics of *S. typhimurium* PipB2 and SteA T3Es in HeLa cells using fluorescence microscopy. By genetically fusing PipB2 and SteA at their C-terminus to GFP11 in conjunction with plasmid-based expression of its complementary fragment (GFP1–10) in the host, it was shown that PipB2 localizes to the tubular network and that SteA co-localizes with a trans-Golgi marker [31], observations in line with their respective reported localizations [49,50]. However, PipB2-GFP11 effector translocation was only detected at four hours post infection (hpi), whilst immunoblotting of native PipB2 showed translocation evidence already after two hpi [31,51]. This discrepancy was attributed to the slow maturation kinetics of split-GFP assembly as fluorescence complementation was shown to take up to ~15–30 min [52]. Consequently, split-GFP is not suited for precise real-time monitoring of effector translocation within the host cell. Further of consideration is that GFP or split-GFP fluorescence requires molecular oxygen, confining its use to aerobic systems [53,54] and its usage is additionally restricted to non-acidic cell compartments as its fluorescence drops significantly at pH levels below 6 (a drop in fluorescence of ca. 50% at pH 6 and ca. 90% at pH 5) [55,56,57]. This implies that for example quantification studies of *S. typhimurium* effectors residing in the SCV is less suited considering the low and variable pH of such SCVs (pH < 4.5) making the use of other fluorophores desirable. For instance, a pH-insensitive GFP variant, e.g., pH-stable tandem dimer GFP (pH-tdGFP), could be considered (Figure 2a) [58].

An alternative fluorescence readout for monitoring effector delivery in the host based on Cre-mediated recombination was reported when studying *S. typhimurium* SopE translocation into COS-2 cells by means of flow cytometry [33]. Here, effectors are fused to the Cre recombinase (343 AAs, 38 kDa). Upon translocation of this effector-Cre fusion into host cells, transfected with a GFP reporter system, Cre-mediated *loxP* recombination results in the expression of GFP caused by the excision of a *LoxP*-flanked terminator sequence located upstream of the GFP coding sequence (CDS). Importantly, translational fusion of Cre recombinase to SopE did not influence Cre recombinase activity–indicative of the proper refolding of Cre upon translocation–nor inhibited SopE translocation. Although this method does not pose the effector-GFP fusion problem, it is impeded by the lack of spatio-temporal information.

As extensively reviewed in [28], fluorescent chaperones provide another elegant alternative method to circumvent the FP fusion problem. In this case, effectors are not tagged, but instead, fluorescent chaperone reporters are expressed inside host cells. Upon effector translocation, these effector-specific chaperones are recruited to the translocated effector, revealing its subcellular localization [59]. A major limitation of this method is the fact that it requires (a priori knowledge on availability of) effector-specific chaperones.

#### 2.1.2. Light-Oxygen-Voltage Sensing (LOV) Domain or Tetracysteine (4Cys) Effector Fusions

Based on a domain originally found in the blue-light photoreceptor phototropin, translational fusions with the bacterial effector- light-oxygen-voltage sensing (LOV) protein domain were shown to provide an alternative manner to study bacterial effector translocation as demonstrated for the pathogenic *E. coli* Tir, *S. typhimurium* SipA and *Shigella flexneri* (*S. flexneri*) IpaB T3Es [34,35]. Upon translocation of effector-LOV fusions into host cells, these LOV domains emit fluorescence as a result of binding to endogenous blue-light sensitive flavin mononucleotide (FMN) chromophores from the host cell. As a result of its smaller size (12 kDa), wider pH range (no fluorescence perturbation between pH 4.5–9.5), compatibility with use in anaerobic environments and rapid maturation kinetics, this LOV system provides several advantages over the (split-)GFP system [60]. However its limited sensitivity (relatively low quantum yield compared to GFP) and near-UV excitation of the fluorophore linked to phototoxicity during imaging refrains its more general use (Table 2) [61].

Another approach for fluorescence-based detection of effectors is the FlAsH (fluorescein arsenical hairpin binder) labeling system. FlAsH makes use of effectors fused to a small peptide tag (12–18 AAs) which contains a tetra-cysteine motif (4Cys), i.e., Cys-Cys-X-X-Cys-Cys in which X can be any amino acid, resulting in fluorescent FlAsH upon binding to this 4Cys tag (Table 2) [62,63]. This method allowed studying the real-time kinetics of the *S. flexneri* effectors IpaB and IpaC, and of the *S. typhimurium* effectors SopE2 and SptP, in a spatio-temporal manner by means of spinning disk confocal microscopy (SDCM) [36,37]. More specifically, when observing real-time translocation making use of FlAsH/4Cys labeling, endogenously tagged IpaB and IpaC T3Es were shown to be rapidly translocated following host cell contact (50% translocation within ca. 4 min) where they interact with actin-rich components [36]. The antagonistic *S. typhimurium* SopE2 and SptP T3Es on the other hand were translocated with different kinetics in line with their function in activating and suppressing Cdc42 (i.e., by showing an enhanced and delayed secretion rate corresponding with the time for invasion and ensuing cellular responses, respectively). These results indicate, that the 4Cys-FlAsH technique provides a powerful mean to study the (fast) kinetics of T3E translocation in real-time into live mammalian cells, enabling to uncover novel spatio-temporal effector regulatory mechanisms [37].

#### 2.1.3. FAST, a New Fluorescent Reporter Allowing Real-Time Monitoring of Effector Translocation

Recent studies on bacterial effector translocation reported the use of fluorogen-activating proteins (FAPs) fusions. In contrast to FPs but similar to the LOV protein domain, FAPs only become fluorescent upon binding to a specific fluorogenic ligand, commonly named fluorogen, substances with very weak fluorescence in their unbound state [64,65]. Since specific binding to the corresponding FAP is required to obtain fluorescence, background fluorescence levels remain very low even in the presence of excess of fluorogen. For example, FAPs coupled to single-chain antibodies were shown to become fluorescent upon fluorogen binding [66,67]. More recently, the Fluorescence-Activating and absorption-Shifting Tag (FAST) was designed and proven to be successful in studying bacterial secretion and translocation [38,68]. Prototypical FAST is a relatively small (14 kDa) protein tag that was designed by means of directed evolution from the Photoactive Yellow Protein (PYP) of *Halorhodospira halophila*. More specifically, the loop that gates the entrance of the fluorogen binding pocket (AA 94–101) was randomized by saturation mutagenesis and clones exhibiting bright fluorescence in the presence of fluorogen were selected. FAST fluoresces upon binding to fluorogens derived from 4-hydroxybenzylidene-rhodanine (HBR) as a result of two spectroscopic changes, namely: (1) an increase of fluorescence quantum yield due to immobilization of the fluorogen and (2) an absorption red shift caused by deprotonation of the fluorogen after binding to FAST (Figure 3a–c) [68]. Importantly, binding of fluorogen to FAST results in good brightness, although slightly lower compared to GFP (Table 2). Furthermore, it was demonstrated that fluorogens bind specific and fully reversible to FAST, and that these fluorogens were shown to be non-toxic in mammalian cells (i.e., HeLa cells) up to concentrations as high as 20 µM, a 2 to 4-fold higher concentration compared to the typically used concentrations in these assays [68,69,70].

To date, several variants of FAST and fluorogens exist and various combinations of their usage result in different fluorescence properties. Inserting a single mutation in the fluorogen binding pocket of FAST resulted for instance in the generation of improved FAST (iFAST), exhibiting slightly decreased dissociation constants for the fluorogens and for some of the fluorogens an increased quantum yield (Table 2) [71]. Dimerization of FAST or iFAST, respectively called tandem FAST (td-FAST) or tandem iFAST (td-iFAST), even led to 2.8-fold and 3.8-fold brighter fluorescence compared to FAST respectively [71]. In addition, td-iFAST was shown to be 1.6-fold brighter than eGFP [71]. Also a slightly smaller variant of FAST–rather misleadingly termed nanoFAST (98AA)–was generated by truncating the first 26 N-terminal residues of FAST since these are not involved in fluorogen binding and are unstructured in the absence of fluorogen [72]. Next to iFAST and nanoFAST, also far-red FAST (frFAST), redFAST and greenFAST were developed, permitting far-red, orange-red and green fluorescence readouts, respectively (Table 2) [73,74]. Remarkably, promiscuous FAST (pFAST) was designed displaying the ability to cover the entire visible spectrum as a result of its capacity to bind various fluorogenic chromophores, each resulting in different fluorescence readouts (Table 2) [75].

As mentioned, in addition to different variants of FAST, also various forms of fluorogens exists such as, for example, HBR-3,5DM (4-hydroxy-3,5-dimethylbenzylidene rhodanine), HBR-3,5DOM (4-hydroxy-3,5-dimethoxybenzylidene rhodanine), and HPAR-3OM (4-hydroxy-3-methoxy-phenylallylidene rhodanine), differing in their emission wavelengths, a property which can be exploited to increase experimental versatility and combined permit multiplex detection (Figure 3d and Table 2) [68,69,73]. For nanoFAST, a new fluorogen called HBR-DOM2 ((4-hydroxy-2,5-dimethoxybenzylidene)-2-thioxothiazolidin-4-one) had to be designed as the conventional fluorogens were unsuited due to the enlarged fluorogen binding pocket of nanoFAST compared to prototypical FAST [72]. The more bulky substituents in HBR-DOM2 likely compensate for this by filling the pocket, thereby making nanoFAST:HBR-DOM2 fluorescent again [72].

While the HBR fluorogens discussed so far are able to permeate both eukaryotic and bacterial membranes [68,76], a new non-permeant fluorogen, i.e., HBRAA-3E, was designed by tethering a carboxymethyl group at the rhodamine head of HMBR, thereby introducing a negative membrane repelling charge at physiological pH (Figure 3d) [70]. This fluorogen allowed the selective labeling of eukaryotic cell-surface FAST-tagged proteins upon transient expression in HeLa cells [70]. Moreover, labeling of these cell-surface proteins was complete within 10 s, indicating rapid labeling kinetics of the FAST reporter system. The capacity of HBRAA-3E to permeate bacterial membranes was also tested by Chekli and colleagues. By making use of an *E. coli* strain expressing periplasmic or cytoplasmic FAST(-fusions), fluorescence was solely detected in the presence of the permeant fluorogen HBR-3,5DM and not in the presence of HBRAA-3E indicating that HBRAA-3E did not pass the outer or inner membrane [76]. Moreover, the utility of this FAST:HBRAA-3E reporter system for prokaryotic labeling of surface proteins was demonstrated by the efficient detection of gram-negative *E. coli* besides gram-positive *L. monocytogenes* cell surface proteins [76].

All the aforementioned advantages and the versatility of FAST made this reporter system an ideal candidate to explore its use in real-time imaging of secreted and translocated bacterial (effector) proteins, as recently tested by Peron-Cane and colleagues [38]. In brief, the use of FAST for Sec- and T3SS-dependent secretion was examined by studying the protein pore-forming toxin listeriolysin O (LLO) from *L. monocytogenes* [80] and the *S. flexneri* effectors OspF and IpaB, respectively. Either full length LLO or the secretion signal peptide (SP) of LLO were N-terminally fused to FAST (SP-FAST). On the other hand, the T3Es OspF and IpaB were C-terminally fused to FAST and expressed from plasmids [38]. Both LLO-FAST and SP-FAST as well as OspF-FAST and IpaB-FAST could be detected in bacterial culture supernatants in the presence of fluorogen by measuring fluorescence, indicating that FAST-tagged proteins allow Sec- or T3SS-dependent secretion respectively and that FAST retains its capability to become fluorescent upon binding its fluorogen after translocation. The versatility of this system was then pushed one step further by real-time detection of protein secretion and effector translocation in the context of infection using spinning disk confocal microscopy (SDCM), overall demonstrating that the FAST tag allows (real-time) monitoring of bacterial effector secretion and dynamics, at least for T3SS- or Sec-dependent secreted proteins.

### 2.2. Enzymatic and Luminescence-Based Readouts to Track Bacterial Effector Translocation

#### 2.2.1. Enzyme-Based Methods: Effector Fusion with Adenylate Cyclase (CyaA), β-Lactamase or R Protein-Mediated Hypersensitive Responses in Planta

Dhrekopf and colleagues described a method to study T3E translocation into plant cells, which makes use of the AvrBs3 effector encoded by *Xanthomonas campestris* [81]. As this transcription activator-like (TAL) effector induces in planta target gene expression, this property was successfully exploited to monitor T3E translocation into plants by C-terminally fusing the effector under study to a derivative of AvrBs3, i.e., AvrBs3∆2, which only encodes the TAL domain. Upon effector translocation, AvrBs3∆2 will activate transcription of the plant Bs3 gene encoding a flavin monooxygenase that upon expression triggers a clearly visible hypersensitive response [81]. Roden et al. used a similar approach making use of an AvrBs2 elicited hypersensitive response to identify new *Xanthomonas campestris* T3Es, i.e., *Xanthomonas* outer proteins (XOPs) [82]. Furthermore, this plant resistance (R) protein-mediated effector recognition system enabled validation of T3Es predicted by machine-learning approaches [83,84,85].

Effector translocation can also be monitored by means of enzyme tags that require a host cell component to become active. Translocation of *S. typhimurium* SteA or *Yersinia* YopE into host cells was for example demonstrated by fusing these effectors to the *Bordetella pertussis* adenylate cyclase (CyaA) protein (1225 AAs, 126 kDa) [39,86]. The N-terminal adenyl cyclase domain of CyaA induces a calmodulin-dependent increase in cAMP upon translocation, which can easily be monitored by means of cAMP detection (e.g., using an anti-cAMP antibody-based ELISA as readout). Additionally, this CyaA reporter system was used to study *Pseudomonas syringae* T3E (i.e., AvrPto and AvrB) and *Ralstonia solanacearum* T3E (i.e., Rip proteins) translocation into plant cells [40,41]. While CyaA has the remarkable intrinsic capacity to translocate its cyclase domain directly across the eukaryotic host cell membrane, this method however does not provide spatio-temporal, nor real-time information on effector translocation.

Another enzyme-based method that relies on a host-derived component for read-out makes use of glycogen synthase kinase (GSK) recognition motif as tag (13 AAs). Upon translocation into the host cell, this tag will become phosphorylated by host kinases, which can subsequently be detected by means of phospho-specific antibodies. This method was successfully used to monitor T3SS-dependent translocation of *Yersinia* Yop proteins as well as T4SS-dependent translocation of *H. pylori* CagA [87].

C-terminal effector-TEM1 β-lactamase (286 AAs, 32 kDa) fusions on the other hand allowed the analysis of *Legionella pneumophila* T4E and Enteropathogenic *E. coli* (EPEC) T3E translocation [88,89]. This system relies on the lipophilic esterified coumarin cephalosporin fluorescein (i.e., CCF2/4-AM) substrate that can easily permeate eukaryotic cell membranes. However, once inside eukaryotic cells, cellular esterases convert it to negatively charged CCF2/4 retained in the cell. As a result of CCF2/4 cleavage by translocated effector-β-lactamase fusions, a switch from green (520 nm) to blue (447 nm) fluorescence indicates effector translocation [89]. Although this method provides a high sensitivity, it cannot be used for effector localization studies within the host cell since cleaved CCF2/4 diffuses throughout the cell. Moreover, real-time studies have to be interpreted with care as the kinetics of CCF2/4 cleavage should be taken into account and CCF2/4 substrate levels may become depleted [42].

Since the TAL, CyaA and β-lactamase based reporter systems are not suited to determine T3E localizations inside the host cell, the use of self-labeling enzymes (SLE) tags, in combination with an appropriate ligand coupled to tetramethylrhodamine (TMR), was explored to enable the spatio-temporal study of T3E translocation inside host cells in real-time, performed by means of super-resolution microscopy thereby providing a spatial resolution of 25 nm (Figure 4) [44]. In this study, the HaloTag, SNAP-tag and CLIP-tag were C-terminally fused to several *S. typhimurium* SPI-1 and SPI-2 effectors and to the YopM T3E of *Yersinia enterolitica*. Translocation and functionality upon translocation into host cells was tested and intriguingly while Halo-tagged SPI-1 effectors (SipA, SopB and SopE) were poorly translocated, SNAP- or CLIP-tagged T3E fusions were translocated efficiently. In contrast, Halo- or SNAP-tagged SPI-2 effectors (PipB2, SifA, SseF and SseJ) were efficiently translocated, whilst CLIP-effector fusions were less efficiently translocated [44]. *Yersinia* YopM on the other hand was only detected when fused to the CLIP-tag. Besides, the use of SLE tags to determine effector localization and real-time dynamics inside host cells was successfully reported for SseF-HaloTag [44]. In a similar context, super-resolution microscopy was also used to study SopB-mEos3.2 fusion protein localization [90]. It should be noted however, that in contrast to the CyaA, GSK and β-lactamase methods, SLE labeling is not restricted to eukaryotic models since it can be performed in bacterial cells as well. Indeed, background signals originating from a pool of non-translocated SNAP-tag labeled PipB2 could be observed [44]. Accordingly, as the effector fusions were expressed from their native chromosomal localization, further investigation is required to unravel whether this is due to partial translocation and thus impairment of effector translocation.

#### 2.2.2. The Use of NanoLuc Luminescence to Study Effector Translocation in Real-Time

In addition to FPs and FAPs, molecular biologists have also greatly benefited from the use of bioluminescence and more specifically from the use of *Renilla* and Firefly luciferases. A new commercially available luciferase called NanoLuc (19 kDa) has recently been used to study bacterial effector secretion. The luciferase originating from the shrimp *Oplophorus gracilirostris* [91] called OLuc, emits light upon addition of the substrate coelenterazine and is composed of two heterodimers each consisting of a large subunit of 35 kDa and a small subunit of 19 kDa. The luminescent properties of OLuc can solely be attributed to the small subunit (OLuc-19), hinting towards the potential use of OLuc-19 on its own [92]. Because of the compromised stability and poor expression of OLuc-19 however, structural optimization by mutagenesis was required, finally resulting in–again a bit misleading with reference to its the still relatively substantial size–NanoLuc exhibiting an 81,000× improved brightness over OLuc-19 [91]. Additional optimization of the substrate leading to the development of furimazine resulted in an even 2.5 million brighter bioluminescent system relative to OLuc-19, and a 150-fold brighter system relative to *Renilla* and Firefly luciferase (Table 3) [91]. Importantly, in the presence of furimazine, NanoLuc luminescence is ATP-independent but O_2_-dependent (Figure 5a). Beside reduced size and improved brightness, NanoLuc offers several advantages over traditional luciferases. First, Nanoluc has an increased thermal stability [91]. Second, the Nanoluc bioluminescence system is not readily affected by pH since it retains its complete activity at pH 7–9 and a significant activity can be observed at pH levels between 5 and 7 (ca. a drop of 50% at pH 6 and 75% at pH 5), while Firefly luciferase cannot readily be used at pH levels below 8 [91]. One disadvantage, however, is the relative short furimazine half-life of 2 h at room temperature, complicating real-time monitoring of effector secretion for longer time periods. Interestingly two types of NanoLuc-based complementation (NanoBiT) systems were designed (Figure 5b) [93]. Herein, a large fragment of 18 kDa, referred to as Large BiT or LgBiT, complements with a small 1.3 kDa fragment of only 11 amino acids comprising one single β strand, called Small BiT or SmBiT, resulting in luminescence. SmBiT does not readily interact with LgBiT due to its low affinity (K_D_ = 190 µM) but is readily suitable to study protein-protein interactions. Indeed, only upon interaction between two proteins under study, either fused to SmBiT or LgBiT, both moieties are physically brought together resulting in NanoLuc-based complementation and luminescence [93]. However, additionally, a high-affinity variant (K_D_ = 0.7 nM) of SmBiT, named HiBiT, is capable to bind LgBiT spontaneously and instantaneously, making this system ideal for protein expression profiling and protein translocation studies.

Besides the study of protein-protein interactions and protein expression analysis, NanoLuc and NanoLuc-based complementation (NanoBiT) systems have also been used to study protein stability among other applications [94]. More recently, the use of NanoLuc bioluminescence to study bacterial effector secretion was exploited. Here, *S. typhimurium* SPI-1 T3E SipA and SopE were C-terminally fused to NanoLuc harboring a Myc tag, and both fusion proteins were expressed in transformed wild type or T3SS-deficient *S. typhimurium*. Expression and secretion profiles of SipA-NanoLuc and SopE-NanoLuc were monitored by Myc-immunodetection probing bacterial cell lysates or filtered culture supernatants and luminescence measurement revealed NanoLuc activity of secreted SipA- and SopE-NanoLuc fusions. The signal to noise ratio (S/N), determined as the ratio of luminescence of control over T3SS-deficient *S. typhimurium*, was significantly higher when making use of Nanoluc fusions compared to fusions to traditional luciferases such as *Renilla* luciferase [43]. These results suggest that the NanoLuc system provides a powerful means to assess T3E secretion. However, of note, secretion of the T3SS substrate SctP–implicated in the regulation of needle length–appeared to be compromised when fused to NanoLuc, again pointing to specific effector-fusion combinations affecting secretion and translocation. This caveat could however be overcome by making use of the split-NanoLuc system in which secretion of the SctP-HiBiT fusion could successfully be detected by complementation with recombinant LgBiT [43]. Furthermore, SipA-NanoLuc and NanoLuc-based complementation of injected SipA-HiBiT in stable HeLa cells expressing LgBiT enabled monitoring of the specific injection (kinetics) of SipA in infected host cells. Here, translocation of SipA-NanoLuc was confirmed by luminescence readout of infected HeLa cells after removal of attached bacteria (the latter needed to avoid potential interference of bacterial background signal due to the permeability of the NanoLuc substrate used), while the translocation dependent complementation of split-NanoLuc only considers host-injected SipA-HiBiT, eliminating the need for removal of extracellular bacteria and permitting the study of T3E injection kinetics.

Next to T3E secretion, the utility of NanoLuc or split-NanoLuc was also demonstrated for Sec-dependent secretion and type IV secretion [95,96,97]. For instance, Sec-dependent translocation of the *E. coli* proteins OmpA and Spy was studied in vitro by making use of Split-NanoLuc and proteoliposomes (PLs), or bacterial inner-membrane vesicles (IMVs) [95]. PLs and IMVs facilitate protein translocation assays relying on the principle that translocated proteins are protected inside these PLs or IMVs and consequently become resistant to proteolysis allowing their detection (e.g., immune- or autoradiographic detection). By adding purified LgBiT into the PL reconstitution mixture–and thus including LgBiT inside PLs or IMVs–or alternatively by making use of LgBiT harboring an inner membrane lipid anchor sequence, translocation of OmpA-HiBiT and Spy-HiBiT was confirmed. Background signals, caused by leakage of LgBiT from PLs or IMVs, were significantly reduced by addition of a catalytically inactive recombinant variant of LgBiT called DrkBiT [95]. Similarly, the split-NanoLuc system enabled monitoring of *Helicobacter pylori* CagA effector translocation by the Cag T4SS [97]. Fusing HiBiT to the N-terminus of CagA and stably expressing the complementary fragment LgBiT in a human gastric adenocarcinoma cell line allowed quantification of translocated CagA, revealing that only a limited amount of CagA is actually translocated into its host. Additionally, real-time monitoring of CagA secretion allowed to study its translocation kinetics demonstrating that CagA is translocated within a few minutes after host cell contact [97].

### 2.3. Genetic Code Expension as a Mean to Study Effector Translocation

The previously described methods to study bacterial effector secretion all have the disadvantage that they require tagging of the effector of interest, which might (partially) interfere with effector secretion or functionality. As a tag-independent approach, genetic code expansion in which a non-canonical amino acid is introduced into the effector sequence of interest can be considered [98]. The non-canonical amino acid must be encoded by a codon that does not already encode for any of the 20 natural occurring amino acids. Therefore, a stop codon (usually TAG) or an artificial four- instead of three-base codon is typically used [99]. A specific designed and expressed tRNA/aminoacyl tRNA synthetase pair (orthogonal tRNA/synthetase) recognizes this stop or artificial codon, resulting in the (limited) co-translational incorporation of the non-canonical amino acid, usually provided in the medium, into the genetically modified target under study (Figure 6a). Intuitively, one would expect that any TAG codon within the genome could be subjected to (residual) non-canonical amino acid incorporation leading to labeling and in case of the use of a stop codon, stop-codon read-through of other proteins besides the protein of interest. However, mutating all TAG codons to TAA stop codons resulted in similar levels of non-specific labeling as wildtype *E. coli* strains [100]. In part, this observation could be linked to the fact that the incorporation efficiency was shown to largely depend on the codon context with AAT-TAG-ACT being the most efficient one [101]. Since *S. typhimurium* T3Es do not harbor such a sequence, non-specific incorporation of non-canonical amino acids in at least *S. typhimurium* T3Es is expected to be minimal [101].

More recently, genetic code expansion was successfully used for the labeling and visualization of the *S. typhimurium* SL1344 SPI-2 encoded T3E SifA and the T3SS component SsaP [98]. SifA plays a key role in the formation of *Salmonella*-induced filaments (SIFs), which are assumed to bring intravacuolar *Salmonella* in contact with other host cell compartments and to deliver endocytosed nutrients to the SCV in order to promote bacterial replication [102]. Previous attempts to tag SifA at its N- or C-terminus were unsuccessful since effector translocation and function was hampered. Inserting two internal HA tags between amino acid D136 and I137 (NCBI Accession Q56061) resulted in functionally active SifA that could be translocated into host cells [103]. C-terminal mCherry labeling of SsaP–involved in the control of SPI-2 effector secretion as substrate specificity switch–was also unsuccessful as cleavage of the fusion protein was observed impeding SsaP imaging [98].

Singh and colleagues however successfully tracked SifA and SsaP translocation and localizations in infected host cells by making use of genetic code expansion [98]. While SifA was labeled with an azide-containing amino acid (i.e., azidophenylalanine (AzF)) at position 52–a site selected because of its surface accessibility and its low conservation–SsaP was labeled with trans-cyclooctene (TCO) lysine corresponding to a TAG codon in a sequence context enabling higher labeling efficiency. Here, the incorporated AzF is able to react with dibenzocyclooctyne (DBCO) resulting in fluorescence. TCO, on the other hand, reacts with tetrazine (Tz)-coupled dyes generating a fluorescent signal only after successful labeling (Figure 6b,c). For both SifA and SsaP, specificity of labeling was confirmed and labeling did not interfere with effector functionality or secretion. Visualization of SifA within SIFs was observed by its colocalization with Lysosomal-associated membrane protein 1 (LAMP-1), a marker of SIFs as well as the motor protein kinesin-1 and SseJ, a SPI-2 T3E involved in SIF formation [98].

## 3. Discussion

Translocation of bacterial effectors into target host cells serves as one of the hallmarks of bacteria-host interactions. Although the structure and function of the T3SS and several other bacterial secretion systems have been studied extensively, less is known about effector translocation dynamics. Consequently, various methods to quantitatively and/or qualitatively monitor effector secretion and translocation (in real-time) have been developed [27,28,38,43,98]. Except for the use of genetic code expansion, these methods require fusion of the effector of interest to a reporter tag. Commonly, effectors are fused at their C-terminus since T3SS-recognition signals and chaperone-binding domains are usually contained within the unstructured N-terminus of effectors. Nevertheless, the incorporation of tags might still hamper (specific) effector translocation or alternatively alter the effector’s localization or function inside the host. Intuitively, one would expect that the protein’s ability to become secreted is largely dependent on the size of the used tag. However, while large structural tags are more likely to cause steric interference compared to small tags, the use of relatively large ‘harmless’ tags has also been reported [104]. This is for instance illustrated by the fact that T3E fusion to CyaA (126 kDa), β-lactamase (29 kDa) or Cre-recombinase (38 kDa) allows efficient translocation, while the smaller GFP tag (27 kDa) impairs translocation [8,18]. Indeed, the thermodynamic and mechanical stability of GFP rather than its size were reported to block T3E secretion [46]. From the recent additions of tags to the toolset (i.e., FAST and NanoLuc), secretion and translocation assays indicate the large non-perturbing nature of these tags, but effector-specific effects of tag fusions have however been reported and cannot generally be excluded, as is the case for possible downstream effects on effector stability, localization and protein interactions. Moreover, fusion of the aforementioned tags to an effector of interest did not render the tags dysfunctional as a result of protein unfolding required for translocation when passing through the T3SS injection needle. Indeed, translocated Cre recombinase, CyaA and β-lactamase retain their enzymatic activity, split-GFP, LOV and FAST their fluorescence, and NanoLuc its luminescence [28,38,43].

While the use of genetic code expansion to monitor bacterial effector secretion has the advantage of not requiring translational tagging [98], the low overall labeling efficiencies, besides non-specific incorporation of non-canonical amino acids have all been recognized [100,101]. Moreover, genetic code expansion also requires expression of tRNA/aminoacyl tRNA synthetase pair which may impose a cellular metabolic burden, thereby potentially compromising the infection potential of the bacterium under study. It is also important to consider that the aforementioned methods all rely on the ability to genetically manipulate the bacterium under study [105]. This is for example complicated or even impossible in case of symbionts, multidrug resistant strains–as antibiotics are typically used for selection–or bacteria that are difficult to transform. The split-GFP11, Cre recombinase-based and NanoLuc complementation methods to study effector translocation additionally require manipulation of host cells to introduce GFP1–10, a *LoxP*-GFP reporter system and LgBiT expression respectively, which is then again limited to cells that can be genetically modified, transfected or transduced.

Another important consideration when selecting the appropriate methodology to study bacterial effector translocation is the requirement of additional substrates/fluorogens required and their associated properties. The addition of substrates or fluorogens is not required when the split-GFP system is used to study effector dynamics. In addition, no substrates are needed when the effector of interest is fused CyaA which relies on the endogenous secondary messenger cAMP expressed by the host [34,39]. Consequently, this implicates that monitoring effector translocation into host cells does not suffer from bacterial background signal, eliminating the need for bacterial removal as was required to study SipA-NanoLuc injection into HeLa cells viewing the presumed bacterial membrane permeability of the NanoLuc furimazine substrate [43]. A problem that could be overcome by making use of the commercially available cell impermeable NanoLuc substrate in analogy with what was carried out for the FAST reporter system [70]. Alternatively, one could make use of translocation-dependent complementation of split-NanoLuc instead [43]. Importantly, monitoring of effector translocation for a longer period of time was made possible with the development of endurazine and vivazine since these substrates have a longer lifetime (up to 72 h) compared to furimazine (2 h) viewing their steady conversion to furimazine by the slow rate of ester hydrolysis catalyzed by cellular esterases, though suffer from lower sensitivity and complicating detection of early translocation events. This problem also applies for the β-lactamase reporter system as a result of rapid CCF2/4 depletion (already after one hour) [42], whilst it does not apply for FAST, as fluorogens can be added in excess to the cells since they only become fluorescent upon binding to FAST, limiting background fluorescence [68]. As a related note, since toxicity effects of substrates or fluorogens have been reported, their use should be tested and optimized before use in the model system under study. FlAsH had for example no influence on bacterial growth or viability at 4-fold higher concentrations (20 µM) than required for performing cellular assays (5 µM). However, FlAsH was shown to be toxic for eukaryotic cells, and increasing concentrations (up to 20 µM) were shown to alter *S. flexneri* bacterial internalization and actin foci formation indicating that optimalization of the FlAsH concentration is required to avoid labeling artefacts whilst still maintaining fluorescence sensitivity [36]. Additionally, it was shown that the FAST fluorogens are not toxic for mammalian cells at 2-fold higher concentrations than normally used in fluorescence assays [68]. Thus far, bacterial toxicity was however not reported but our unpublished data indicates no growth perturbation making use of the generally used concentration (5 µM), a finding in contrast to the bacterial growth inhibition observed when making use of furimazine concentrations recommended for use in mammalian assays, indicating the additional importance of assessing bacterial besides host cell toxicity.

The development of the split-NanoLuc reporter system provided a new approach–next to LOV, 4Cys/FlAsH and β-lactamase (Table 1)–to monitor effector translocation in real-time [43]. It would be interesting to assess whether the same principles can be applied to the FAST reporter system as until now effector secretion was only measured at fixed timepoints [38]. Accordingly, bacteria with an endogenously FAST-tagged effector could be cultivated in the presence of a membrane impermeable fluorogen. Moreover, the FAST tag could be used in conjunction with both a cell permeable and impermeable fluorogen for the detection of bacterial effector expression and quantification of secreted or translocated effector concentrations. In case that the FAST-tag would interfere with the secretion of specific effectors, a split-FAST system could be considered (Figure 1). Split-FAST was previously used for real-time visualization of protein-protein interactions by fusing the bait or prey to the N-terminal fragment (AA 1–114) of FAST (called NFAST) or to the C-terminal part (AA 115–125) of FAST (called CFAST) [106]. Here, the effector should preferably be fused to the smaller CFAST fragment, while larger NFAST could be added recombinantly to the medium to study secretion, or alternatively, expressed in the host when studying effector translocation in accordance to the split-NanoLuc system.

It is important to mention that most of the bacterial translocation studies report on observations of effectors translocated with high abundance such as the *S. typhimurium* T3Es PipB2, SteAT3E, etc., since they are not sensitive enough to monitor effectors at low abundance [107]. In this context, the use of tandem tags such as trimeric GFP11 or 4Cys tags were found to slightly improve sensitivity of detection (e.g., 15% increase for 3x-4Cys-SptP; 3-fold increase for 3x-GFP11-SteA) [37,108]. Genetic code expansion however was shown to enable the visualization of lowly abundant effectors once the low non-canonical amino acid incorporation has been overcome [98]. Furthermore, it is quite striking that about 90% of the reported studies on bacterial effector translocation tools make use of only very few well-studied bacteria such as *S. typhimurium*, *S. flexneri*, etc., implicating that further research is required to obtain a more comprehensive understanding of bacterial effector translocation.

Interestingly, the use of multimodal tags, originating from the use of the tandem affinity purification (TAP) tag, became more common [104]. The future use of non-perturbing multimodal tags could be explored for studying bacterial effector biology as combining translocation dynamics together with purification or interactomics approaches could provide novel insights on an effector’s function. For instance, the additional inclusion of a BioID tag (e.g., BirA*) for conducting proximity-dependent biotin labeling to enable the capture of interacting proteins–might allow the study of effector-host protein-protein interactions at selected timepoints during infection [109].

In this review, we discussed previously reported besides more recently developed state-of-the-art methods for elucidating (real-time) bacterial effector secretion, and elaborated on their latest advances and shortcomings. Implementation of these methods will enlighten an effector’s mode of action by providing knowledge of timing of effector secretion, quantities of secreted effector, order of effector secretion and context-dependent effector network dynamics [110,111]. Integrating these data with for example effector-host protein interaction data or effector expression data (e.g., dual RNA-seq data) will eventually enable us to shed better lights on bacterial effector biology, potentially leading to the development of new therapeutics [112].

## Figures and Tables

**Figure 1 microorganisms-10-00260-f001:**
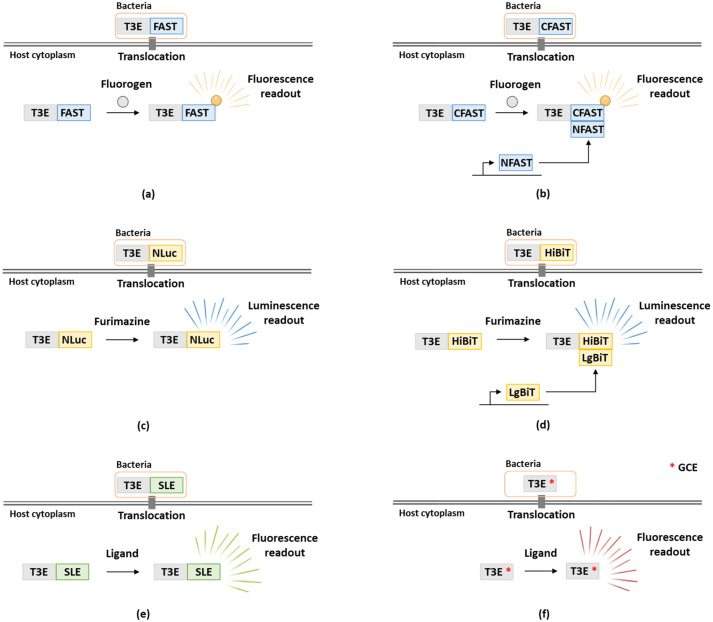
Expansion to the toolset to study bacterial effector translocation. (**a**) The FAST-based method was recently developed to monitor bacterial effector translocation. To this end, bacterial effectors are fused to the FAST tag. Upon translocation and addition of fluorogen, fluorescence can be measured. (**b**) Hypothetically, a split-FAST system can also be used. Here, the shorter C-terminal fragment of FAST (CFAST) is fused to the effector of interest, whilst the complementing N-terminal fragment (NFAST) is expressed inside host cells. Upon translocation, complementation takes place, resulting in fluorescence in the presence of fluorogen. (**c**) The use of NanoLuc (NLuc) to study bacterial effector translocation. Upon translocation of the effector fused to NLuc and addition of furimazine, luminescence is measured. (**d**) Similar to split-FAST, a split-NanoLuc complementation-based system (NanoBiT) allows real-time monitoring of effector translocation. In this case, HiBiT is fused to the effector of interest and LgBiT is expressed inside host cells. (**e**) The use of self-labeling enzyme (SLE) tags (e.g., HaloTag, CLIP or SNAP) to monitor effector translocation upon addition of the appropriate ligand. (**f**) Using genetic code expansion (GCE), a non-canonical amino acid (indicated with a red *) is incorporated into the effector under study, allowing monitoring of translocation inside host cells.

**Figure 2 microorganisms-10-00260-f002:**
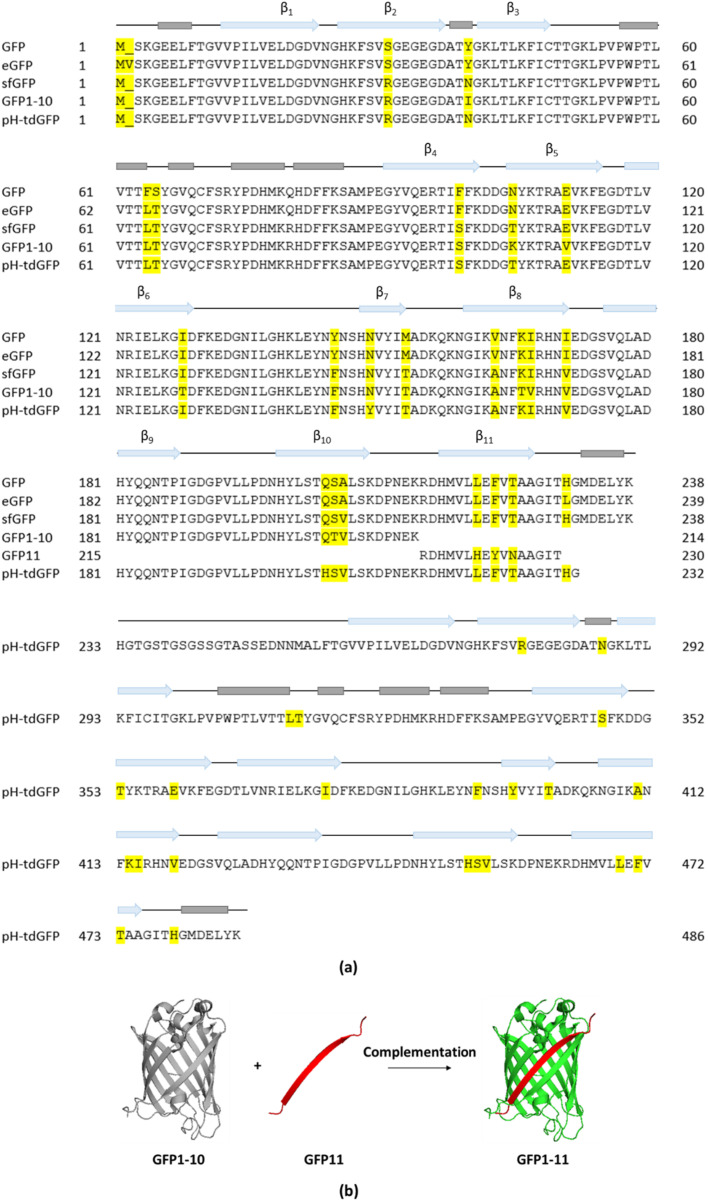
(**a**) Sequence alignment and structural annotation of GFP, eGFP, sfGFP, splitGFP and pH-tdGFP. Helices are indicated in grey and β-strands as blue arrows. Differing amino acid identities in the alignments are highlighted in yellow. (**b**) Split-GFP complementation: separating the 10 N-terminal β-strands (GFP1–10) from the 11th β-strand (GFP11) results in the generation of split-GFP that becomes fluorescent upon complementation (Created in PyMOL).

**Figure 3 microorganisms-10-00260-f003:**
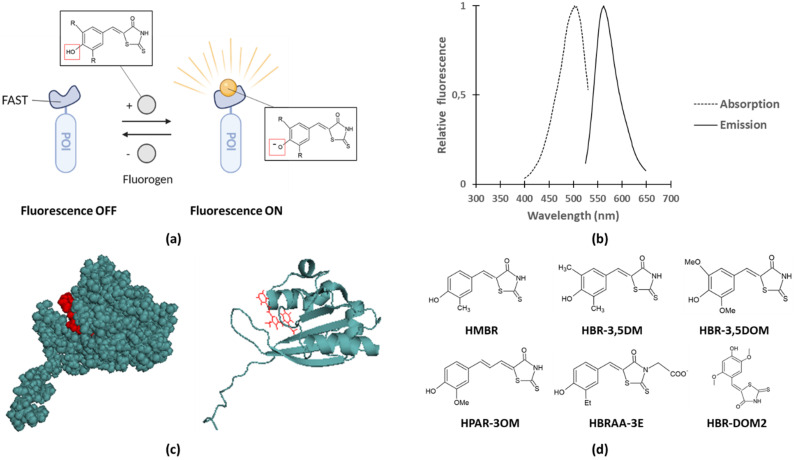
FAST as a reporter system. (**a**) FAST fluoresces upon binding to a fluorogen, enabling fluorescent detection of a protein of interest (POI). Fluorescence is a result of an increase in quantum yield and an absorption red shift caused by deprotonation of the fluorogen after binding to FAST. (**b**) Absorption and emission spectra of FAST-bound HBR-3,5 DM. 5 µM HBR-3,5 DM was added to recombinantly produced FAST (10 µM) in a total volume of 100 µL PBS (pH 7.4), and fluorescence recording performed at room temperature using a Spark 10M multimode microplate reader (Tecan). (**c**) Protein structure of FAST in complex with N871b as chromophore-ligand as solved by Mineev and colleagues (PDB: 7AVA) [72]. Upon FAST-binding, N871b displays optical spectra resembling HBR-DOM fluorogen binding. Figure was created using PyMOL. (**d**) Chemical structures of different fluorogens.

**Figure 4 microorganisms-10-00260-f004:**
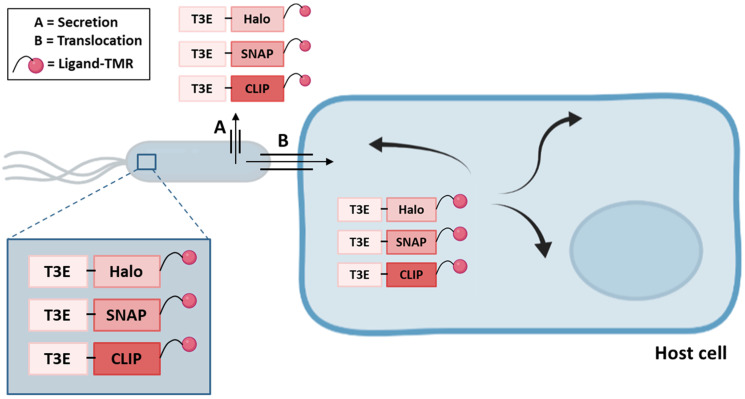
The use of SLE tags to monitor bacterial effector secretion or translocation. SLE-tagged effector fusions (i.e., HaloTag, SNAP-tag or CLIP-tag) are expressed in bacteria. Ligand-TMR (bacterial and host cell membrane permeable) addition permits the labeling of bacteria or infected host cells when tracking expressed/secreted or translocated T3E-SLE fusions, respectively. Upon translocation into host cells, the dynamics and subcellular localization of the T3Es under study can be monitored.

**Figure 5 microorganisms-10-00260-f005:**
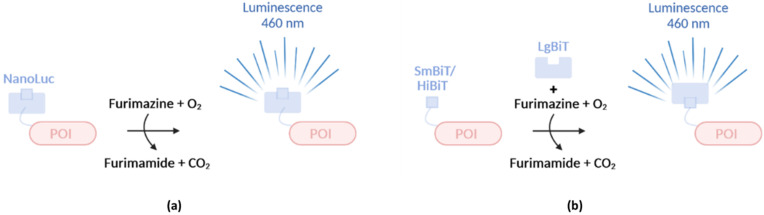
NanoLuc and NanoLuc-based complementation systems. (**a**) NanoLuc, fused to a protein of interest (POI), emits light (460 nm) upon addition of its substrate Furimazine in an O_2_-dependent manner. (**b**) Two types of NanoLuc-based (NanoBiT) complementation systems were developed. Herein, a large fragment called LgBiT complements with a smaller HiBiT/SmBiT peptide resulting in luminescence.

**Figure 6 microorganisms-10-00260-f006:**
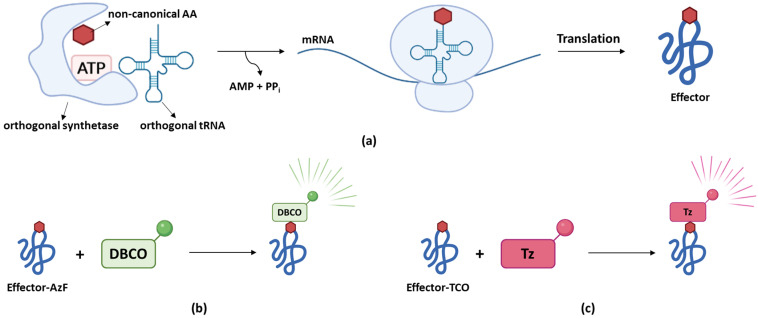
Genetic code expansion. (**a**) Schematic representation of the introduction of a non-canonical amino acid into the effector under study. A specifically designed orthogonal synthetase charges (in an ATP-dependent manner) the orthogonal tRNA with a non-canonical amino acid, which subsequently integrates this non-canonical amino acid within the nascent polypeptide chain translated by the ribosome. Following translation, the effector has the non-canonical amino acid incorporated in its sequence. (**b**) Effectors containing AzF are able to react with DBCO resulting in fluorescence. (**c**) Incorporation of TCO into the effector under study allows labeling with Tz-coupled dyes generating fluorescence upon labeling.

**Table 1 microorganisms-10-00260-t001:** Overview of methods to monitor bacterial effector translocation. * indicates protein family members. Abbreviations used; FC: flow cytometry, FMN: Flavin mononucleotide, PR: plate reader, SRM: super-resolution microscopy, TMR: tetramethylrhodamine, N: N-terminal and C: C-terminal.

Tag	Counterpart	Readout	Size of Tag (kDa)	Trans-Location	Host Sub-Cellular Localization	Prok./Euk. Differentiation	Real-Time	Representative Effectors Studied	Tag Position	Ref.
Split-GFP: GFP11	GFP1–10	Fluorescence (microscopy)	27	✓	✓	✓	-	*S. typhimurium* PipB2, SteA	C	[31]
								*P. syringae* AvrB, AvrRps4	C	[32]
Cre-lox	LoxP-GFP reporter	Fluorescence (FC)	38	✓	-	✓	-	*S. typhimurium* SopE	C	[33]
LOV	FMN chormophore (Host)	Fluorescence microscopy/SDCM	10	✓	✓	✓	✓	*E. coli* Tir	C	[34]
								*S. typhimurium* SipA	C	[35]
								*S*. *flexneri* IpaB	C	[34]
4Cys	FlAsH	Fluorescence (SDCM)	<1	✓	✓	✓	✓	*S. flexneri* IpaB/C	C	[36]
								*S. typhimurium* SopE2, SptP	C	[37]
FAST	Fluorogen	Fluorescence (SDCM)	14	✓	✓	✓	✓	*S. flexneri* OspF, IpaB	C	[38]
CyaA	Calmodulin (Host)	ELISA	126	✓	-	✓	-	*Yesrsina pseudoturberculosis* YopE	N	[39]
								*P. syringae* AvrPto, AvrB, Hop *	C	[40]
								*R. solanacearum* Rip *	C	[41]
β-lactamase	CCF2	Fluorescence (PR)	29	✓	-	✓	✓	EPEC Tir, Map, EspF, EspG	C	[42]
NanoLuc	Furimazine	Luminescence (PR)	19	✓	-	-	✓	*S. typhimurium* SipA, SopE	C	[43]
NanoBiT	Furimazine	Luminescence (PR)		✓	-	✓	✓	*S. typhimurium* SipA	C	[43]
Self-labeling enzymes	Ligand (e.g., ligand-TMR)	Fluorescence (SRM)	20–33	✓	✓	-	✓	*Yersinia pestis* YopM	C	[44]
								*S. typhimurium* SPI-1 and SPI-2 T3E	C	[44]

**Table 2 microorganisms-10-00260-t002:** Overview of physicochemical properties of the fluorescent reporter systems relying on the FPs GFP, mCherry, or alternatively on LOV:FMN, FlAsH:4Cys or various FAST:fluorogen combinations.

Fluorescent Reporter System	Absorption (nm)	Emission (nm)	Φ (%)	ε (mM^−1^ cm^−1^)	K_D_ (µM)	Ref.
eGFP	488	507	60	56	/	/
mCherry	587	610	22	72	/	[77]
LOV:FMN	300–500	450–495	20–40	13	/	[61,78]
4Cys:FlAsH	508	528	49	30–80	10^−5^	[62,79]
FAST:HMBR	481	540	23	45	0.13	[68]
FAST:HBR-3,5DM	499	562	49	48	0.08	[69]
FAST:HBR-3,5DOM	518	600	31	39	0.97	[69]
FAST:HBRAA-3E	505	559	8	61	1.30	[70]
iFAST:HMBR	480	541	23	41	0.07	[71]
iFAST:HBR-3,5DM	499	558	57	46	0.06	[71]
iFAST:HBR-3,5DOM	516	600	40	38	0.41	[71]
nanoFAST:HBR-DOM2	502	563	56	26	0.85	[72]
FAST:HBR-DOM2	510	566	54	31	0.021	[72]
frFAST:HMBR	484	550	13	56	0.50	[73]
frFAST:HBR-3,5DOM	525	600	19	51	3.90	[73]
frFAST:HPAR-3OM	555	670	21	45	1.00	[73]
greenFAST:HMBR	478	544	23	40	0.09	[74]
redFAST:HBR-3,5DOM	556	603	29	43	1.20	[74]
pFAST:HMBR	481	542	23	54	0.01	[75]
pFAST:HBR-3,5DM	501	561	44	49	0.01	[75]
pFAST:HBR-3,5DOM	520	600	35	44	0.06	[75]
pFAST:HBRAA-3E	506	558	5	53	0.05	[75]

**Table 3 microorganisms-10-00260-t003:** Overview of different luciferases, their substrates and properties.

Luciferase	Substrate	Size (kDa)	Emission (nm)	Brightness	Ref.
FLuc	Coelenterazine	61	565	/	[91]
RLuc	Coelenterazine	36	480	0.5	[91]
OLuc-19	Coelenterazine	19	460	0.00009	[91]
NanoLuc	Coelenterazine	19	460	2.4	[91]
NanoLuc	Furimazine	19	460	76	[91]

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
