# Peer review of "Recent Advancements in Tracking Bacterial Effector Protein Translocation"

_microorganisms, 2022, doi:10.3390/microorganisms10020260_

Round 1
Reviewer 1 Report
The review by Braet et al. describes the techniques that have been used to monitor effector protein secretion in the T3SS. It is a comprehensive review and will be a good contribution to the field.
Minor changes:
Ln 38: Please change nine to ten secretion systems. See ref Palmer, T 2021 Mol Microbiol.
Ln 308: Two full stops
Ln 342: Please add reference.
Ln 324, Ln 371, Ln 492: These sentences do not make sense. Please fix it.
Author Response
Please see the attachment entitled "Responses to comments - reviewer 1".

Reviewer 2 Report
The review submitted by Braet and colleagues summarizes the scope of methods utilized for monitoring bacterial effector protein secretion and provides some insights into the future of monitoring said secretion. The review is well written and well organized. The topic is of high interest to researchers working in the field of secretion systems and this provides a nice (relatively short) summation of the content in the field. There are, however, several locations within the manuscript where where substitution of specific examples for vague wording, providing more specific examples, and/or including illustrative figures would significantly strengthen the manuscript. The following comments/suggestions should be considered prior to publication:
1) Line 77-78: It would be nice if the authors provided more specific detail with respect to the consensus sequences that are discussed as "indicative of protein secretion signals"
2) The authors provide a nice summary of slit GFP and its uses, but it would help immensely if they provided a structure illustrating the components and how they assemble.
3) While the main point of the review is to discuss the approaches themselves, it would be beneficial to provide some context with respect to what was ultimately learned in the studies that are provide as examples. For example, lines 202-208 discuss the use of the FlAsH labeling system to monitor effector secretion in real time. What were the findings? What were the rates that were observed? How do these rates compare to those found via other methods?
4) Figure 3: Including a structure of the FAST fluorogen activating protein component would be a helpful illustration. The absorption/emission spectra could be removed is space is needed.
5) Lines 322-324. The authors state, "Indeed, as a result of coumarin cephalosporin fluorescein (CCF2) cleavage by translocated β-lactamase, a switch from green (520 nm) to blue (323nm) fluorescence could be observed in case of infected HeLa cells were stained [78]." The meaning of this statement is unclear, perhaps specifically due the the latter part of the sentence stating, "in case of infected HeLa cells were stained".
6) I am uncertain if it is possible due to copyright, but including a figure illustrating the localization of effectors as described in the section discussing self labeling enzymes (lines 329-342) would make a good addition and an attractive figure demonstrating the power of the method.
7) Lines 371-372. It is unclear why the low affinity interaction between SmBiT and LgBiT makes it "suitable for studying protein protein interactions". Either revisit this statement or provide an explanation of this statement in the paper.
8) Lines 373-375. The authors state, "In contrast to SmBiT, this HiBiT fragment is capable to bind LgBiT spontaneously, making this system ideal for protein expression profiling and protein translocation studies." I do not understand how a Kd is cited for the interaction between SmBiT and LgBiT (190uM) yet its interaction is now said to not be spontaneous...
9) line 536: It would be helpful if the authors provided values in reference to FlAsH being cytotoxic toward eukaryotic cells. What levels were toxic? What concentrations are required for an experiment? The same information would be helpful (as a comparison) for the FAST system discussed in the same paragraph.
Author Response
Please see the attachment entitled "Responses to comments - reviewer 2".

Reviewer 3 Report
-The authors made a very elaborate pros and cons list for each translocation system in the discussion but placed relatively less focus on the ability of each reporter to differentiate between eukaryotic and prokaryotic cells, which is very critical for translocation studies. Additionally they did not place an emphasis on the necessary facilities that are required for each system (confocal microscopy, fluorescence/luminescence readers, specific cell lines…). Considering this is a method review this information should mentioned as well.
-Even though the manuscript briefly describes in the introduction that effector translocation is required for plant-bacteria interactions, it does not mention anything about the large body of work dedicated for assessing bacterial translocation into the plant cells which also includes the CyaA and split-GFP reporters used in mammalian cells (doi.org/10.1105/tpc.17.00047, doi: 10.1128/JB.186.2.543-555.2004, doi.org/10.1094/MPMI-23-3-0251). Outside of these reporters, plants systems harbor one of the oldest and most user-friendly binary translocation systems that is based on C-terminal fusion of the HR-domain of avirulence effectors which results in rapid localized cell death in crops harboring the corresponding resistance gene to the avirulence reporter (doi: 10.1007/978-1-4939-6649-3_11) . Considering that this system was successfully used as a screening method in many studies as a validation tool for effector prediction (doi.org/10.1073/pnas.0407383101, doi:10.1094 /MPMI -22-11-1401, doi: 10.1111/mpp.12288, doi: 10.1111/mpp.12528, doi: 10.1111/mpp.12877), at least an honorable mentioning is in order.
-Table 1: CyaA was used as a C-terminal tag for T3SS effectors as well as an N-terminal tag for T4SS effectors (doi.org/10.1105/tpc.17.00047, doi: 10.1128/JB.186.2.543-555.2004, doi.org/10.1094/MPMI-23-3-0251).
-Text references does not match citation numbers. For instance, reference 89 from the text should be 86, reference 80 from the text should be 78 and the list goes on. Please make sure that the reference list is in order before publication.
-Some of the citations does not contain the year of publication. Please correct this issue.
-Please properly use italicized and upper case fonts in the manuscript: for instance, in lines 132, 140, 152, 166, 387 and others "S. Typhimurium" should be changed to "S. typhimurium"
-Figure 2 – resolution is poor. Please improve picture quality.
-The authors should provide some brief introduction regarding the location of the secretion signal in the target proteins of each secretion systems and its relevance to the type of translational fusions of the described reporters. For instance, T4SS effectors should be fused to an N-terminal reporter since the secretion signal is found in the C-terminus while sec and T3SS effectors should be fused to a C-terminal reporter since the secretion signals are found in the N-terminus.
-320-328 – beta-lactamases are active also outside of eukaryotic cells. The use of a CCF2/4-AM, which is esterified within eukaryotic cells as a substrate, is required for this system. This should be clearly mentioned din the text.
-329-342 – Unlike CyaA, GSK and beta-lactamase (with the right substrate) SLE labeling is not specific to eukaryotic cells and can be also observed in the bacterial cells as well. This should be mentioned in the text.
-329-342 – citation is missing. Please add reference number "38" (Göser et al. 2019).
-430-371- this paragraph does not explain how genetic code expansion labeling actually works. There is no mentioning of enzymatic reaction of the labeled amino acid with a substrate as elaborated in the Singh manuscript (Singh et al. 2021, figure 1). Considering the fact that this is a novel method, I would expect a more elaborated description would be added to this section.
Author Response
Please see the attachment entitled "Responses to comments - reviewer 3".

Round 2
Reviewer 3 Report
The authors addressed all the issues that were raised in the first review cycle. I personally find the review very informative and think it will serve as a great resources for many groups that are interested in effector biology
One small correction:
Line 532: The translocation studies described in ref 82-85 used a similar but not an identical approach to the TAL-based reporter system. In all studies, validation of translocation is based on elicitation of hypersensitive response through immune recognition by a plant resistance gene/protein. However, AvrBs2 and AvrBs1 are not TAL effectors and their corresponding resistance gene is not activated by the same mechanism as AvrBs3. In the case of AvrBs2, recognition is based on the presence of the intracellular NB-LRR receptor Bs2 (https://doi.org/10.1073/pnas.96.24.14153).
Therefore, it is not correct to classify these reporters as "TAL-based reporter system" and something in the line of "R protein mediated recognition "is more suitable.
Author Response
We thank the reviewer for the kind appreciation of our work and the extra valuable comment made. Accordingly, we adapted (the title of) the paragraph to the following: ‘Drehkopf and colleagues described a method to study T3E translocation into plant cells, which makes use of the AvrBs3 effector encoded by Xanthomonas campestris [1]. As this transcription activator-like (TAL) effector induces in planta target gene expression, this property was successfully exploited to monitor T3E translocation into plants by C-terminally fusing the effector under study to a derivative of AvrBs3, i.e. AvrBs3∆2, which only encodes the TAL domain. Upon effector translocation, AvrBs3∆2 will activate transcription of the plant Bs3 gene encoding a flavin monooxygenase that upon expression triggers a clearly visible hypersensitive response [1]. Roden et al. used a similar approach making use of AvrBs2 elicited hypersensitive responses to identify new Xanthomonas campestris T3Es, i.e. Xanthomonas outer proteins (XOPs) [2]. Furthermore, this plant resistance (R) protein-mediated effector recognition system enabled validation of T3Es predicted by machine-learning approaches [3–5].’
References
- Drehkopf, S.; Hausner, J.; Jordan, M.; Scheibner, F.; Bonas, U.; Büttner, D. A TAL-Based Reporter Assay for Monitoring Type III-Dependent Protein Translocation in Xanthomonas. In Type 3 Secretion Systems: Methods and Protocols; Nilles, M.L., Condry, D.L.J., Eds.; Methods in Molecular Biology; Springer: New York, NY, 2017; pp. 121–139 ISBN 978-1-4939-6649-3.
- Roden, J.A.; Belt, B.; Ross, J.B.; Tachibana, T.; Vargas, J.; Mudgett, M.B. A Genetic Screen to Isolate Type III Effectors Translocated into Pepper Cells during Xanthomonas Infection. PNAS 2004, 101, 16624–16629, doi:10.1073/pnas.0407383101.
- Teper, D.; Burstein, D.; Salomon, D.; Gershovitz, M.; Pupko, T.; Sessa, G. Identification of Novel X Anthomonas Euvesicatoria Type III Effector Proteins by a Machine‐learning Approach. Mol Plant Pathol 2015, 17, 398–411, doi:10.1111/mpp.12288.
- Nissan, G.; Gershovits, M.; Morozov, M.; Chalupowicz, L.; Sessa, G.; Manulis-Sasson, S.; Barash, I.; Pupko, T. Revealing the Inventory of Type III Effectors in Pantoea Agglomerans Gall-Forming Pathovars Using Draft Genome Sequences and a Machine-Learning Approach. Molecular Plant Pathology 2018, 19, 381–392, doi:10.1111/mpp.12528.
- Jiménez-Guerrero, I.; Pérez-Montaño, F.; Da Silva, G.M.; Wagner, N.; Shkedy, D.; Zhao, M.; Pizarro, L.; Bar, M.; Walcott, R.; Sessa, G.; et al. Show Me Your Secret(Ed) Weapons: A Multifaceted Approach Reveals a Wide Arsenal of Type III-Secreted Effectors in the Cucurbit Pathogenic Bacterium Acidovorax Citrulli and Novel Effectors in the Acidovorax Genus. Molecular Plant Pathology 2020, 21, 17–37, doi:10.1111/mpp.12877.